# User-defined outcomes of the Danish cardiovascular screening (DANCAVAS) trial: A post hoc analyses of a population-based, randomised controlled trial

**Axel Cosmus Pyndt Diederichsen**[1]*, Anna Mejldal[2], Rikke Søgaard[3], Jesper Hallas[4], Jess Lambrechtsen[5], Flemming Hald Steffensen[6], Lars Frost[7], Kenneth Egstrup[5], Martin Busk[6], Grazina Urbonaviciene[7], Marek Karon[8], Lars Melholt Rasmussen[9], Jes Sanddal Lindholt[10]

1 Department of Cardiology, Odense University Hospital, Odense, Denmark, 2 Open Patient Data Explorative Network, Department of Clinical Research, Odense University Hospital, Odense, Denmark, 3 Elite Research Centre for Individualised Medicine, Odense University Hospital, Odense, Denmark, 4 Department of Clinical Pharmacology, University of Southern Denmark, Odense, Denmark, 5 Department of Cardiology, Svendborg Hospital, Svendborg, Denmark, 6 Department of Cardiology, Lillebaelt Hospital, Vejle, Denmark, 7 Department of Cardiology, Regional Hospital Central Jutland, Silkeborg, Denmark, 8 Department of Medicine, Nykøbing Falster Hospital, Nykøbing Falster, Denmark, 9 Department of Clinical Biochemistry and Pharmacology, Odense University Hospital, Odense, Denmark, 10 Department of Cardiothoracic and Vascular Surgery, Odense University Hospital, Odense, Denmark

* axel.diederichsen@rsyd.dk

**Data Availability Statement:** Data cannot be shared publicly because of Danish data protection rules. Data are available from the Danish Statistics (https://www.dst.dk/en/informationsservice/informationsspecialister) for researchers who meet the criteria for access to confidential data.

## Abstract

### Background

The Danish cardiovascular screening (DANCAVAS) trial, a nationwide trial designed to investigate the impact of cardiovascular screening in men, did not decrease all-cause mortality, an outcome decided by the investigators. However, the target group may have varied preferences. In this study, we aimed to evaluate whether men aged 65 to 74 years requested a CT-based cardiovascular screening examination and to assess its impact on outcomes determined by their preferences.

### Methods and findings

This is a post hoc study of the randomised DANCAVAS trial. All men 65 to 74 years of age residing in specific areas of Denmark were randomised (1:2) to invitation-to-screening (16,736 men, of which 10,471 underwent screening) or usual-care (29,790 men). The examination included among others a non-contrast CT scan (to assess the coronary artery calcium score and aortic aneurysms). Positive findings prompted preventive treatment with atorvastatin, aspirin, and surveillance/surgical evaluation. The usual-care group remained unaware of the trial and the assignments. The user-defined outcome was based on patient preferences and determined through a survey sent in January 2023 to a random sample of 9,095 men from the target group, with a 68.0% response rate (6,182 respondents). Safety outcomes included severe bleeding and mortality within 30 days after cardiovascular

**Funding:** This work was supported by:—The Southern Region of Denmark (No grant number, to JL),—The Danish Heart Foundation (16-R107-A6671, to JL),—The Danish Independent Research Councils (4183-00174, to JL). The funders of the study had no role in the study design, data collection and analysis, decision to publish, or preparation of the manuscript.

**Competing interests:** The authors have declared that no competing interests exist.

**Abbreviations:** CAC, coronary artery calcium; CI, confidence interval; CT, computer tomography; DANCAVAS, Danish cardiovascular screening; HU, Hounsfield unit; HR, hazard ratio; IPTW, inverse probability of treatment weighting; IQR, interquartile range; MACE, major adverse cardiovascular events; MALE, major adverse limb events; NNI, number needed to invite.

surgery. Analyses were performed on an intention-to-screen basis. Prevention of stroke and myocardial infarction was the primary motivation for participating in the screening examination. After a median follow-up of 6.4 years, 1,800 of 16,736 men (10.8%) in the invited-to-screening group and 3,420 of 29,790 (11.5%) in the usual-care group experienced an event (hazard ratio (HR), 0.93 (95% confidence interval (CI), 0.88 to 0.98; $p = 0.010$); number needed to invite at 6 years, 148 (95% CI, 80 to 986)). A total of 324 men (1.9%) in the invited-to-screening group and 491 (1.7%) in the usual-care group had an intracranial bleeding (HR, 1.17; 95% CI, 1.02 to 1.35; $p = 0.029$). Additionally, 994 (5.9%) in the invited-to-screening group and 1,722 (5.8%) in the usual-care group experienced severe gastrointestinal bleeding (HR, 1.02; 95% CI, 0.95 to 1.11; $p = 0.583$). No differences were found in mortality after cardiovascular surgery. The primary limitation of the study is that exclusive enrolment of men aged 65 to 74 renders the findings non-generalisable to women or men of other age groups.

## Conclusion

In this comprehensive population-based cardiovascular screening and intervention program, we observed a reduction in the user-defined outcome, stroke and myocardial infarction, but entail a small increased risk of intracranial bleeding.

## Trial registration

ISRCTN Registry number, ISRCTN12157806 https://www.isrctn.com/ISRCTN12157806.

## Author summary

### Why was this study done?

- Stroke and heart attack remain a prevalent cause of decreased quality of life and premature death.

- Coronary atherosclerosis serves as an important risk modifier and is easily identifiable by cardiac imaging.

- Despite this, screening for cardiovascular disease by cardiac CT in the Danish cardiovascular screening (DANCAVAS) trial did not lead to a decrease in all-cause mortality.

- However, it is essential to recognise that patients may have preferences other than death, and these preferences play a critically important role in informed and shared decision-making between caregivers and patients.

### What did the researchers do and find?

- The DANCAVAS trial included men aged 65 to 74 from specific areas of Denmark, with some receiving invitations to screening (16,736 men) and others receiving usual care (29,790 men).

- Patients with significant atherosclerosis were treated with atorvastatin, aspirin, and underwent surveillance or surgery in case of aneurysms.

- Here, the study outcomes were determined based on patient preferences, which were assessed through a survey of a random sample of 9,095 men aged 65 to 74. The surveyed men expressed a preference for preventing stroke and heart attack over death.

- Screening reduced the absolute risk of stroke and heart attack by 0.7%, but it also increased the risk of intracranial haemorrhage by 0.2%.

## What do these findings mean?

- The DANCAVAS trial met patient preferences, but the benefits were modest and accompanied by a slight increase in the risk of intracranial haemorrhage.

- These data suggest that there is limited benefit to the widespread implementation of cardiovascular screening programmes.

- The trial specifically targeted men aged 65 to 74, meaning the findings cannot be generalised to women or men outside this age group.

- Future cardiovascular screening programs should encompass both sexes and accommodate patient preferences. Additionally, they might consider excluding aspirin from the intervention arm.

## Introduction

Cardiovascular diseases remain the most frequent cause of death in adults living in middle- and high-income countries and are associated with a reduction in disability-adjusted life years [1]. A systematic review from Cochrane suggests that health checks have little or no effect on ischemic heart disease, stroke, and mortality [2]. The use of imaging to quantify coronary artery calcification (CAC) score is emerging as a new way of identifying patients with increased risk as the images reflect the presence of early disease [3]. The Danish cardiovascular screening (DANCAVAS) trial is, to our knowledge, the first population-based randomised clinical trial including CAC score in risk stratification. Men aged 65 to 74 years were invited to a comprehensive, advanced cardiovascular screening examination, or usual care. The screening did not result in a decrease in all-cause mortality, although a clinically important benefit could not be ruled out [4].

In primary prevention, it may no longer be sufficient to focus only on all-cause mortality, as morbidity and treatment costs are increasing. In addition, it is noteworthy that preferences to avoid death compared to avoid nonfatal events appears to be highly age dependent. Individuals below 65 years of age weigh avoiding death highest, while individuals above 65 years of age prioritise avoiding myocardial infarction or stroke higher than death [5]. Thus, patient preferences are critical important for well-informed shared decision-making.

In the DANCAVAS trial, the primary outcome, all-cause mortality, was defined without involvement from the target group. In this study, we aimed to assess the community interest in

cardiovascular screening, user preferred outcomes for such screening, and the effects of screening on these outcomes.

## Methods

### Trial design

This is a post hoc analysis of the DANCAVAS trial. Here, we evaluate the outcome as defined by the target group. DANCAVAS is a multicentre, parallel-group, unmasked randomised controlled cardiovascular screening trial involving all men 65 to 74 years of age who were living in 15 selected municipalities in the southern and central regions of Denmark [6]. Written, informed consent was obtained from all participants who attended the screening program. The trial was conducted with the approval by the Southern Denmark Region Committee on Biomedical Research Ethics (ID S-20140028) and the Danish Data Protection Agency. The statistical analysis plan of this follow-up study was designed prior to collection of data by the first and last author with input from the trial steering committee. The first author has full access to all the data in the study and takes responsibility for its integrity and the data analysis. The trial protocol and statistical analysis plan can be accessed at ISRCTN registry; ISRCTN12157806.

### Randomisation, screening program, and intervention in the DANCAVAS trial

All men—without exceptions—65 to 74 years of age who were living in 15 selected municipalities in the southern and central regions of Denmark were identified in the Danish Registries and included in DANCAVAS. From September 2014 through September 2017, a computer-based algorithm randomised these men 1:2 to invitation-to-screening (16,736 men, of which 10,471 underwent screening) or usual-care (29,790 men) (Fig 1). The randomisation procedure was stratified according to geographic area. Immediately after the randomisation, participants in the invited-to-screening group were invited to the screening examination by a digital mail in e-Boks, an online mailbox used by Denmark's municipalities and state authorities. By definition, participants in the invitation-to-screening group became aware of the trial-group

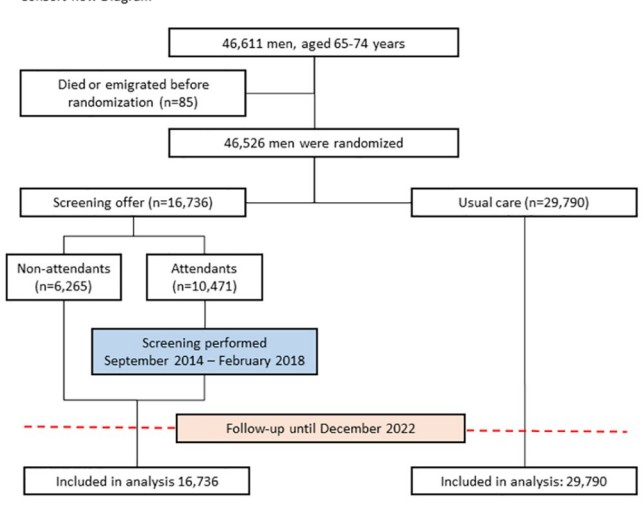

**Fig 1. Enrolment, randomisation, and follow-up.**

**Table 1. Baseline characteristics for the randomised groups.**

| Characteristic | Invited-to-screening (N = 16,736) | Usual-care (N = 29,790) |
|---|---|---|
| Age—mean (standard deviation) | 68.8 (2.6) | 68.8 (2.6) |
| Prescriptions the last year before randomisation | | |
| • Antiplatelet agents—No (%) | 4,184 (25.0%) | 7,596 (25.5%) |
| • Anticoagulants—No (%) | 1,440 (8.6%) | 2,581 (8.7%) |
| • Lipid-lowering agents—No (%) | 6,247 (37.3%) | 11,439 (38.4%) |
| • Antihypertensive agents—No (%) | 8,741 (52.2%) | 15,763 (52.9%) |
| • Antidiabetic agents—No (%) | 2,163 (12.9%) | 3,765 (12.6%) |
| Hospital admission during the last 5 years before randomisation | | |
| • Stroke—No (%) | 748 (4.5%) | 1,521 (5.1%) |
| • Ischaemic heart disease* –No (%) | 637 (3.8%) | 1,227 (4.1%) |
| • Heart failure—No (%) | 396 (2.4%) | 748 (2.5%) |
| • PAD—No (%) | 373 (2.2%) | 643 (2.2%) |
| • Aortic aneurysms—No (%) | 277 (1.7%) | 449 (1.5%) |

* Ischaemic heart disease: myocardial infarction and coronary revascularization.

PAD; peripheral arterial disease.

assignments, while participants in the usual-care group, including their physicians, were not notified about the trial and thus were unaware of the assignment. Treatment in the usual-care group relies on the individual person, but risk assessment, including measurement of blood pressure, lipids, and HgbA1c, at the primary physician is a free and common service. Baseline characteristics are shown in Table 1.

The screening examinations were performed at 4 Danish centres from September 2014 through February 2018. The attendance rate to the screening examination was 62.6% (10,471 of 16,736 men participated). The screening program included a non-contrast cardiac-CT scan to detect CAC, aortic and iliac aneurysms and atrial fibrillation; ankle–brachial blood-pressure measurements to detect peripheral artery disease and hypertension; and a blood sample to detect diabetes mellitus and hypercholesterolemia. The screening method and findings has been described previously [7].

Participants with a CAC score above the median score for sex and age [8], aneurisms or peripheral artery disease were recommended atorvastatin 40 mg and aspirin 75 mg per day. Additionally, participants with aneurisms were, depending on diameter, referred for surveillance or surgery. Participants with atrial fibrillation were referred for cardiac evaluation; while patients with hypertension, diabetes, and hypercholesterolemia were referred to the primary physicians.

## Survey to the target group

The user-defined outcome was determined through a survey to the target group (Fig A in S1 Supplementary Material (Danish) and Fig 2 (English)). Target group included men between 65 and 74 years of age. We identified approx. 10,000 of these men (aiming for 1,000 in each age-group by birth year). The men were randomly selected and not aware of the DANCAVAS trial. The questions were developed together with 8 representatives from the target group. The survey was sent by digital mail (E-boks, see above) in January 2023.

The recipients were asked about their interest in a cardiovascular screening examination. If they reported interest in having a screening examination they were asked, in lay terms, to choose a main reason for their willingness to participate: to prevent stroke, myocardial

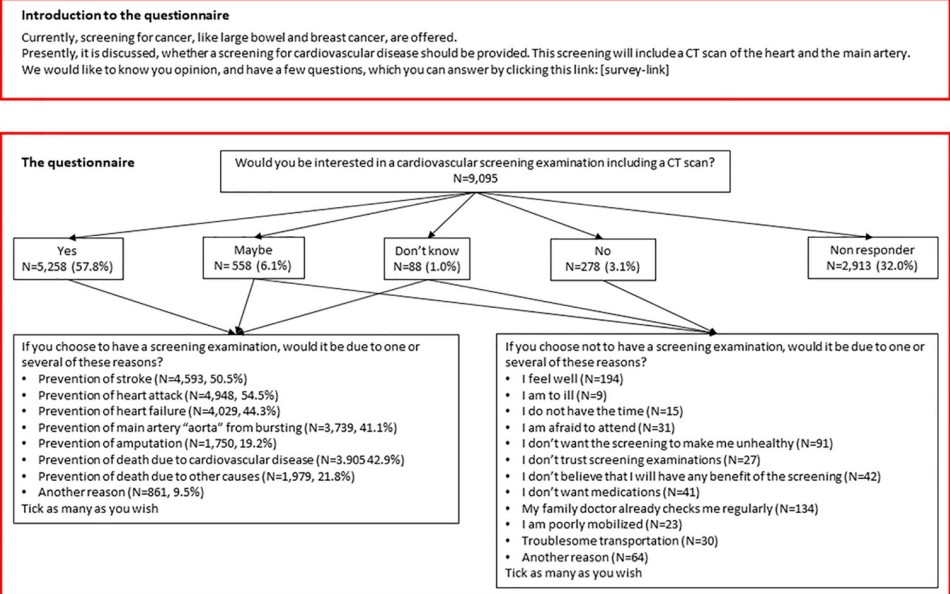

**Fig 2. Survey regarding interest in a cardiovascular screening examination.**

infarction, heart failure, aortic rupture/dissection, limb amputation, death due to cardiovascular disease, and/or death due to any cause. With these answers, the outcome was prespecified in the statistical analysis plan (ISRCTN12157806) before the survey was completed and defined as a composite of reasons chosen by more than 50% of the population. Additionally, if they were not interested in such screening, they were asked why they deselected the screening option.

## Outcome

Respondents in the survey defined the user-defined primary outcome. Secondary outcomes were defined by the investigators. In both cases, the outcome was registry-based (Table A in S1 Supplementary Material). Secondary outcomes were:

- Composite outcome of major adverse cardiovascular events (MACE) (stroke (including ischemic and haemorrhagic), acute myocardial infarction, heart failure, and death due to cardiovascular disease),

- Composite outcome of major adverse limb events (MALE) (aortic dissection and rupture, revascularisation due to critical limb ischemia, major amputation due to peripheral arterial disease, and death due to cardiovascular disease),

- Individual components of the composite secondary outcomes.

Exploratory outcomes were attendance rate, initiation, and adherence to preventive medications and elective aortic aneurysm repair after randomisation.

Safety outcomes were prespecified by the investigators and included potential harms by aspirin (intracranial bleeding and gastrointestinal bleeding leading to hospitalisation) and incident cancer (due to radiation from the cardiac-CT scan) from 6 months after randomisation.

## Data sources

The preferences in the target group were gathered by surveys sent to randomly selected men 65 to 74 years old at the time of posting.

Outcome data were derived from the Danish National Central Person Registry, the Danish National Patient Registry, The Cause of Death Register, and the Danish National Prescription Registry and were assessed at December 31, 2022 (expect cause of death as information was not available after December 31, 2021). All codes are displayed in Table A in S1 Supplementary Material.

## Statistical analysis

All analyses were performed as intention-to-screen and as superiority analyses except a per protocol analysis. The outcomes were compared for the 2 randomisation groups using Cox hazard regression for analyses of unadjusted hazard ratio (HR) (95% confidence intervals (CIs)). Time of randomisation was used to define the onset of risk time, and exit from analysis was time of event or censoring due to death not included as event or censoring on 12-31-2022 whichever came first. Deaths without secondary events were right-censored. Only the first event of each category was counted. Both relative and absolute risk estimates were reported, as well as the number needed to invite (NNI) in order to prevent one primary outcome after 6 years of follow-up, estimated using Newcombe's method [9].

Preventive medications were reported as counts separately for each group and compared between groups by HR (95% CI). Individuals who had received a relevant prescription within 1 year before randomisation were excluded from the analyses.

The model assumption of proportional hazards was assessed with the use of Schoenfeld residuals tests and visual inspection of log-log plots of outcome versus analysis of time.

In addition, to evaluate the impact of the screening examination per se, rather than the impact of invitation-to-screening, we compared the outcomes from probably-attendee-to-screening within the invitation-to-screening and usual-care groups in a per-protocol analysis. Thus, based on participation versus non-participation in the invitation-to-screening group, we calculated the probability of participation. This calculation was based on our knowledge of the baseline characteristics of each person in the invitation-to-screening group. Taking selection bias into account, we used the inverse probability of treatment weighting (IPTW) method [10]. The IPTW method used propensity score to balance the baseline patient characteristics, including age, prior diagnoses, cohabiting status, ethnicity, educational level, income, and employment status, in the invitation-to-screening and usual-care groups by weighting each individual in the analysis by the inverse probability of attending screening. The propensity score $p(X)$ is the conditional probability of attending screening given pre-screening characteristics. The IPTW was calculated as 1 for screened individuals, and $p(X)/(1-p(X))$ for controls, and truncated to the 1st and 99th percentile [11]. For the per-protocol analyses, this IPTW method was added to all subanalyses as described above.

Data management and statistical analyses were performed with Stata 16.1 software from StataCorp.

## Results

### Preferences in the target group

The survey was sent to 9,095 men who were 65 to 74 years of age in January 2023, 6,182 (68.0%) responded, and 5,258 of these were interested in participating in a screening examination, 558 were maybe interested, 88 did not know if they were interested, while 278 were not

**Table 2. Primary and secondary outcomes.**

| | Events No (%) | Years at risk Median (IQR) | No. of events per 100 person-years | Events No (%) | Years at risk Median (IQR) | No. of events per 100 person-years | Incidence risk difference (95% CI) | HR (95% CI) | P value |
|---|---|---|---|---|---|---|---|---|---|
| | Invited-to-screening (N = 16,736) | | | Usual-care group (N = 29,790) | | | | | |
| **Primary outcome** | | | | | | | | | |
| Defined by the users* | 1,800 (10.76) | 6.36 (5.66; 7.08) | 18.19 | 3,420 (11.48) | 6.36 (5.62; 7.08) | 19.61 | −1.42 (−2.49; −0.35) | 0.93 (0.88; 0.98) | 0.010 |
| **Secondary outcomes** | | | | | | | | | |
| MACE | 2,556 (15.27) | 5.37 (4.68; 6.08) | 30.31 | 4,810 (16.15) | 5.36 (4.66; 6.08) | 32.33 | −2.02 (−3.51; −0.53) | 0.94 (0.89; 0.98) | 0.009 |
| MALE | 947 (5.66) | 5.59 (4.84; 6.16) | 10.65 | 1,839 (6.17) | 5.55 (4.74; 6.16) | 11.70 | −1.04 (−1.91; −0.18) | 0.91 (0.84; 0.99) | 0.021 |
| Cardiovascular specific mortality | 633 (3.78) | 5.59 (4.90; 6.18) | 7.06 | 1,175 (3.94) | 5.59 (4.80; 6.16) | 7.40 | −0.34 (−1.04; 0.35) | 0.96 (0.87; 1.05) | 0.353 |
| Stroke | 1,358 (8.11) | 6.37 (5.68; 7.16) | 13.53 | 2,557 (8.58) | 6.37 (5.66; 7.08) | 14.44 | −0.90 (−1.81; 0.01) | 0.94 (0.88; 1.00) | 0.055 |
| Myocardial infarction | 483 (2.89) | 6.40 (5.74; 7.16) | 4.71 | 964 (3.24) | 6.40 (5.68; 7.16) | 5.32 | −0.61 (−1.15; −0.07) | 0.89 (0.79; 0.99) | 0.030 |
| Heart failure | 943 (5.63) | 6.40 (5.74; 7.16) | 9.27 | 1,738 (5.83) | 6.37 (5.68; 7.08) | 9.66 | −0.39 (−1.13; 0.36) | 0.96 (0.89; 1.04) | 0.309 |
| Aortic dissection | 30 (0.18) | 6.55 (5.80; 7.16) | 0.29 | 65 (0.22) | 6.46 (5.74; 7.16) | 0.35 | −0.06 (−0.20; 0.07) | 0.82 (0.53; 1.26) | 0.362 |
| Aortic rupture | 28 (0.17) | 6.55 (5.80; 7.16) | 0.27 | 61 (0.20) | 6.48 (5.74; 7.16) | 0.33 | −0.06 (−0.19; 0.07) | 0.81 (0.52; 1.27) | 0.359 |
| Amputation due to PAD | 96 (0.57) | 6.55 (5.80; 7.16) | 0.92 | 169 (0.57) | 6.43 (5.74; 7.16) | 0.92 | 0.01 (−0.23; 0.24) | 1.01 (0.78; 1.29) | 0.966 |

* The user-defined composite outcome encompassed myocardial infarction or stroke.

Outcomes are registry based (Table A in S1 Supplementary Material).

CI, confidence interval; HR, hazard ratio; IQR, interquartile range; MACE, major adverse cardiovascular events (death due to cardiovascular disease, stroke, acute myocardial infarction, heart failure); MALE, major adverse limb events (death due to cardiovascular disease, aortic dissection and rupture, critical limb ischemia, and major amputation due to peripheral arterial disease); PAD, peripheral arterial disease.

interested. The remaining 2,913 men did not answer the survey. Prevention of stroke and myocardial infarction was selected by 4,593 (50.5%) and 4,948 (54.4%), respectively, as the primary motivation for participating in the screening examination. These responses constituted the user-defined composite outcome (Fig 2). The most common causes to deselect the screening option were "I feel well" (N = 194), "I don't want the screening to make me unhealthy" (N = 91), and "My family doctor already checks me regularly" (N = 134).

## User-defined outcome

The result of the DANCAVAS trial was then analysed by means of the user-defined primary outcome (composite outcome comprising stroke and myocardial infarction). After a median

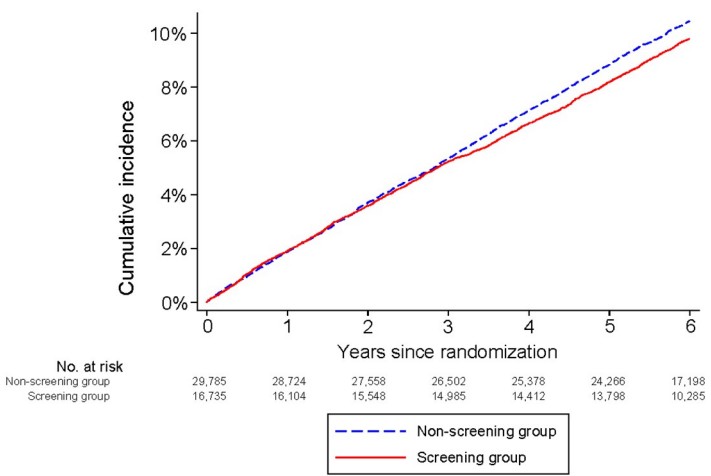

**Fig 3. Cumulative incidence of the user-defined composite outcome (stroke and myocardial infarction).**

follow-up of 6.4 years (IQR 5.7 to 7.1), 1,800 of 16,736 men (10.8%) in the invited-to-screening group and 3,420 of 29,790 (11.5%) in the usual-care group experienced an event; HR, 0.93 (95% CI, 0.88 to 0.98; $p$ = 0.010), corresponding to an NNI at 6 years of 148 (95% CI, 80 to 986) (Table 2 and Fig 3). The relationship between age and the user-defined primary outcome is displayed in Fig B in S1 Supplementary Material. The HR is significantly lower among the younger.

## Secondary outcomes

Results with respect to the secondary outcomes are shown in Table 2. MACE (HR, 0.94; 95% CI, 0.88 to 0.98), MALE (HR, 0.91; 95% CI, 0.84 to 0.99), and myocardial infarction (HR, 0.89; 95% CI, 0.79 to 0.99) were reduced in the invited-to-screening group, while no difference was shown in death due to cardiovascular disease, stroke, aortic or peripheral arterial diseases.

## Per-protocol analysis

The user-defined primary outcome for the probably-attendee-to-screening within the invited-to-screening group compared to the probably-attendee-to-screening within the usual-care group are shown in Table B in S1 Supplementary Material. The risks of the user-defined primary outcome were reduced (HR, 0.87; 95% CI, 0.81 to 0.94). Similar or larger reduced risks were seen in all analysed outcomes. The area under the curve was 0.663 to predict attendance in the invited-to-screening group Fig C in S1 Supplementary Material. Distribution of variables before and after inverse probability of treatment weighting is displayed in Table C in S1 Supplementary Material.

## Exploratory outcomes

Initiation of antiplatelet and lipid-lowering agents was increased in the invited-to-screening group, while there were no differences in initiation of anticoagulants, hypertensive and antidiabetic agents (Table D in S1 Supplementary Material). Likewise, elective aneurysm repair was more common in the invited-to-screening groups (0.9% versus 0.6%; HR, 1.58; 95% CI, 1.26 to 1.97). During follow-up, use of antiplatelet and lipid-lowering agents remained higher in the

**Table 3. Safety outcomes.**

| | Events— No (%) | Years at risk Median (IQR) | No. of events per 1000 person-years | Events— No (%) | Years at risk Median (IQR) | No. of events per 1,000 person-years | Incidence risk difference (95% CI) | HR (95% CI) | *p* Value |
|---|---|---|---|---|---|---|---|---|---|
| | Invited-to-screening (*N* = 16,736) | | | Usual-care group (*N* = 29,790) | | | | | |
| Severe bleeding | 1,296 (7.74) | 6.37 (5.68; 7.16) | 12.87 | 2,177 (7.31) | 6.37 (5.68; 7.08) | 12.20 | 0.67 (−0.19; 1.54) | 1.05 (0.98; 1.13) | 0.130 |
| - Intracranial | 324 (1.94) | 6.55 (5.74; 7.16) | 3.13 | 491 (1.65) | 6.40 (5.74; 7.16) | 2.68 | 0.46 (0.04; 0.87) | 1.17 (1.02; 1.35) | 0.029 |
| - Gastrointestinal | 994 (5.94) | 6.40 (5.68; 7.16) | 9.81 | 1,722 (5.78) | 6.37 (5.68; 7.08) | 9.59 | 0.22 (−0.54; 0.98) | 1.02 (0.95; 1.11) | 0.583 |
| Cancer | 3,857 (23.05) | 6.18 (5.16; 7.05) | 41.48 | 7,033 (23.61) | 6.15 (5.02; 7.05) | 42.70 | −1.22 (−2.87;0.43) | 0.97 (0.93; 1.01) | 0.159 |

Outcomes are registry based (Table A in S1 Supplementary Material).

CI, confidence interval; HR, hazard ratio; IQR, interquartile range.

entire invited-to-screening group compared to the usual-care group (Fig D in S1 Supplementary Material), but adherence was reduced (Table E in S1 Supplementary Material).

## Safety outcomes

Safety outcomes are shown in Table 3. After a median follow-up of 6.4 years, 324 of 16,736 men (1.9%) in the invited-to-screening group and 491 of 29,790 men (1.7%) in the usual-care group had an intracranial bleeding (HR, 1.17; 95% CI, 1.02 to 1.35; *p* = 0.029), while 994 men (5.9%) in the invited-to-screening group and 1,722 (5.8%) the usual-care group had a gastrointestinal bleeding leading to hospitalisation (HR, 1.02; 95% CI, 0.95 to 1.11; *p* = 0.583). No differences were found in incident cancer from 6 months after randomisation.

## Discussion

We evaluated if the comprehensive cardiovascular screening trial, DANCAVAS, would reduce an outcome defined by the users. The user-defined outcome was established based on patient preferences, determined through a survey to the target group. The intervention led to a reduction in the user-defined composite outcome, stroke, and myocardial infarction.

As DANCAVAS investigators, we defined the primary outcome to be all-cause mortality. However, recognising that the target group may have differing preferences, we conducted a survey to ascertain their preferences. The response rate was 68.0%, with the vast majority expressing a keen interest in the screening examination. Notably, the actual participation rate in the DANCAVAS trial was 62.6%. Therefore, we believe that the interest of the respondents and the underlying reasons are both reliable and relevant. The majority of the target group prioritised prevention of stroke and myocardial infarction, while a smaller proportion prioritising the prevention of heart failure, aortic dissection/rupture, amputation due to peripheral artery disease, and cardiovascular death. As older patients assign importance to avoiding myocardial infarction or stroke over mortality, these preferences were not unexpected [5]. Nevertheless, these results emphasise the need of patient (or user) involvement when defining outcomes in clinical trials. While the survey achieved a high response rate, it should be noted

that it did not include arguments for and against cardiovascular screening examination. Thus, the answers may differ if potential benefits and harms were described.

The user-defined primary outcome, stroke and myocardial infarction, was reduced with 7% in the intention-to-screen analysis. As we offered a complex intervention, we do not know the effects of the individual interventions. However, in the invited-to-screening group use of anti-platelet and lipid-lowering agents were increased and elective abdominal aortic aneurysm repair were more common, while there was no difference in prescription of anticoagulants, antihypertensive, and antidiabetics. Notably, there were no more revascularisations in the invited-to-screening group. The curves separates after 3 years, thus the effect might be due to increased use of lipid-lowering agents in the invited-to-screening group, as the effect of anti-platelet treatment is not presumed to be delayed. While the risk of the composite outcome was reduced, we found an increased risk of intracranial bleedings. Opposed to this, there was no differences in severe gastrointestinal bleedings.

These findings are in agreement with recent randomised trials. Prescribing aspirin to middle-aged or elderly persons did not reduce the risk of cardiovascular disease, but entailed a higher risk of major haemorrhage among the elderly [12,13]. However, according to US guideline from 2019, aspirin may be considered for primary prevention in patients 40 to 70 years that are at higher risk of atherosclerotic cardiovascular disease [14], while the European guideline from 2021, state that aspirin may be considered in patients with definitive evidence of coronary artery disease on imaging [15]. Modelling studies from Dallas Heart and Multi-Ethnic Study of Atherosclerosis (MESA) suggest that participants with CAC≥100 Hounsfield Units (HU) have favourable risk/benefit ratios for aspirin use [16,17]. The Nice guidelines from 2023 discuss these findings, but due to lack of randomised trials, aspirin is not recommended despite CAC>100 [18]. In the DANCAVAS trial, we recommended aspirin, if the CAC score was above the median score for sex and age. As the median CAC score in Danish men aged 65 to 74 years is 161 HU, the vast majority of those recommended aspirin had scores above 100 HU [7]. According to the findings in our trial, treatment with aspirin, in agreement with the Nice guidelines, may not be indicated solely based on presence of CAC.

The main analyses were intention-to-screen; however, non-attendance to screening was more than one out of 3 invited, and this may have attenuated the effect. Due to high likelihood of healthy user selection bias, unadjusted comparison of outcomes among those who actually attended the screening and those randomised to usual-care would be highly inappropriate. Instead, we used the IPTW method. With this method, we balanced all available baseline characteristics (including socioeconomic status) in the invited-to-screening and usual-care groups. As expected, the benefits of the screening examination were amplified. However, this attempted per-protocol analysis should be interpreted with caution, as the model to predict attendance evaluated by ROC curve analysis was poor. Causes such as unmatchable better health and higher personal health literacy among attenders may cause better outcomes in the attended-to-screening group and overestimate the benefits seen in the per-protocol analysis.

To our knowledge, this is the first population-based, randomised, cardiovascular screening and intervention trial including CAC score in risk stratification. However, there are several limitations. First, when we planned the DANCAVAS trial in 2013, we were unaware of the importance of user involvement. Consequently, the survey, including the user-defined outcome, was not prespecified. It is crucial to note that, by definition, we as investigators did not define the outcome. Additionally, the statistical analysis plan was completed before data collection. Second, a significant limitation is the exclusive inclusion of men in the trial. Nonetheless, we conducted a pilot study involving 1,044 men and 1,016 women aged 65 to 74 years [19]. Notably, in this pilot study, significant screening findings such as a severely increased CAC score and aortic aneurysms were much lower among women. Consequently, we decided to

focus on men, as the purpose was to identify those with severe imaging-based cardiovascular conditions likely to benefit from preventive treatments. Third, we only included men aged 65 to 74 years, and they were mainly Caucasian. Thus, restricting external validity and generalisability. Fourth, the participation rate was relatively poor, and this may have deteriorated the intention-to-screen analyses. In this manuscript, we attempt to address this through a per-protocol analysis, but this model also has its flaws, as the model to predict attendance was poor. Also, inclusion of men with prior cardiovascular diseases may have impaired the intention-to-screen analyses. Fifth, since the screening and intervention were multifaceted, we are unable to determine which components were beneficial and which were harmful. Sixth, information about outcomes were retrieved from administrative registries, but we consider it unlikely that misclassification of outcomes was dependent on randomisation status.

Based on these results, it seems possible that this comprehensive cardiovascular screening and intervention program would reduce the user-defined outcome, stroke and myocardial infarction, but with an increased risk of intracranial bleedings. Thus, this screening and intervention program is not yet matured to implement to prevent cardiovascular diseases. Future cardiovascular screening programs should encompass both sexes and accommodate patient preferences. Additionally, they might consider excluding aspirin from the intervention arm.

## Supporting information

**S1 Supplementary Material.** Fig A. Survey regarding interest in a cardiovascular screening examination—in Danish. Fig B. Hazard ratio by age of the user-defined primary outcome. Fig C. Resulting ROC curve of the inverse probability of treatment weighting model. Fig D. Initiation of antiplatelet (A) and lipid-lowering (B) agents during follow-up. Table A. Definition of outcomes. Table B. Primary and secondary outcomes for probably attendee within the invitation-to-screening and usual-care group. Table C. Distribution of variables before and after inverse probability of treatment weighting. Table D. Initiation of preventive medication. Table E. Adherence to preventive medications.
(DOCX)

**S1 CONSORT Checklist. CONSORT 2010 checklist of information to include when reporting a randomised trial.**
(DOC)

**S1 Statistical Analysis Plan. Statistical Analysis Plan for the Danish Cardiovascular Screening Trial 2.**
(PDF)

## Author Contributions

**Conceptualization:** Axel Cosmus Pyndt Diederichsen, Lars Melholt Rasmussen, Jes Sanddal Lindholt.

**Data curation:** Axel Cosmus Pyndt Diederichsen, Anna Mejldal, Rikke Søgaard, Jesper Hallas, Jes Sanddal Lindholt.

**Formal analysis:** Axel Cosmus Pyndt Diederichsen, Anna Mejldal, Jes Sanddal Lindholt.

**Funding acquisition:** Axel Cosmus Pyndt Diederichsen, Jes Sanddal Lindholt.

**Investigation:** Axel Cosmus Pyndt Diederichsen, Jess Lambrechtsen, Flemming Hald Steffensen, Lars Frost, Kenneth Egstrup, Martin Busk, Grazina Urbonaviciene, Marek Karon, Lars Melholt Rasmussen, Jes Sanddal Lindholt.

**Project administration:** Axel Cosmus Pyndt Diederichsen, Lars Melholt Rasmussen, Jes Sanddal Lindholt.

**Validation:** Axel Cosmus Pyndt Diederichsen.

**Writing – original draft:** Axel Cosmus Pyndt Diederichsen, Jes Sanddal Lindholt.

**Writing – review & editing:** Axel Cosmus Pyndt Diederichsen, Anna Mejldal, Rikke Søgaard, Jesper Hallas, Jess Lambrechtsen, Flemming Hald Steffensen, Lars Frost, Kenneth Egstrup, Martin Busk, Grazina Urbonaviciene, Marek Karon, Lars Melholt Rasmussen, Jes Sanddal Lindholt.

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
