## [Editor Report · Decision Letter 0]

19 Jan 2024

Dear Dr Diederichsen, 

Thank you for submitting your manuscript entitled "User-defined outcomes of the Danish Cardiovascular Screening (DANCAVAS) trial: a population-based, randomised controlled trial" for consideration by PLOS Medicine.

Your manuscript has now been evaluated by the PLOS Medicine editorial staff and I am writing to let you know that we would like to send your submission out for external peer review.

Please re-submit your manuscript within two working days, i.e. by Jan 23 2024 11:59PM.

Feel free to email me at pdodd@plos.org or the team at plosmedicine@plos.org if you have any queries relating to your submission.

Kind regards,

Pippa

Philippa Dodd, MBBS MRCP PhD

PLOS Medicine

---

## [Decision Letter · Decision Letter 1]

14 Feb 2024

Dear Dr. Diederichsen,

Many thanks for submitting your manuscript "User-defined outcomes of the Danish Cardiovascular Screening (DANCAVAS) trial: a population-based, randomised controlled trial” (PMEDICINE-D-24-00161R1) to PLOS Medicine. The paper has been reviewed by three subject experts and a statistician; their comments are included below and can also be accessed here: [LINK]

As you will see, the reviewers were very positive about the paper, all agreeing that it is innovative in its use of a “user defined” end-point, but raise some points for clarification about specific study details and the methodological approach. After discussing the paper with the editorial team and an academic editor with relevant expertise, I’m pleased to invite you to revise the paper in response to the comments. We plan to send the revised paper to some or all of the original reviewers*, and of course we cannot provide any guarantees at this stage regarding publication.

When you upload your revision, please include a point-by-point response that addresses all of the reviewer and editorial points, indicating the changes made in the manuscript and either an excerpt of the revised text or the location (eg: page and line number) where each change can be found. Please submit a clean version of the paper as the main article file and a version with changes marked should as a marked-up manuscript. Please also check the guidelines for revised papers at http://journals.plos.org/plosmedicine/s/revising-your-manuscript for any that apply to your paper.

We ask that you submit your revision by March 6th 2024. However, if this deadline is not feasible, please contact me by email, and we can discuss a suitable alternative.

Please don’t hesitate to contact me directly with any questions (pdodd@plos.org). If you reply directly to this message, please be sure to ‘Reply All’ so your message comes directly to my inbox.

Kind regards,

Pippa

Philippa Dodd, MBBS MRCP PhD

PLOS Medicine

plosmedicine.org

pdodd@plos.org

*Please note: If your article is accepted, you may have the opportunity to make the peer review history publicly available. The record will include editor decision letters (with reviews) and your responses to reviewer comments. If eligible, we will contact you to opt in or out.

Editorial comments:

1) We really appreciate the premise of your study, and we agree with the reviewers that the “user-defined” outcome is innovative. However, we have some concerns regarding the potential for introducing unintended bias given the approach taken to determine the outcome. Please include a copy of the survey that was sent to the participants to complete as supporting information when you re-submit your manuscript. Further, when considering that the primary trial demonstrated a significant reduction in MACE, we have some concerns regarding novelty of the outcome data presented here. 

2) Please be explicit in the title and the abstract that is a post-hoc sub-study/follow-on study of the primary trial. We suggest reporting in-line with CONSORT explicitly stating the sub-study nature and ensuring that the abstract details the main trial items in 2-3 sentences, including the study population, dates, intervention and primary outcome. The majority of the abstract should then describe the complete details of this sub-study. Please complete the CONSORT checklist and ensure that all components of CONSORT are present in the manuscript as well as clearly defined details of this sub-study. When completing the checklist, please use section and paragraph numbers, rather than page numbers as these often change in the event of publication.

3) We agree with the academic editor (please see below) that there is a lack of clarity regarding how participants were recruited for this sub-study and request that additional details are included. We think that the respondents were an entirely different group of men but please clarify and be explicit in your methods.

4) You refer to ‘prespecified secondary outcomes’ in your methods section including MACE which, as noted above, have already been reported in the primary trial as such we are confused about the use of ‘prespecified’ here. Are these the same secondary outcomes as the detailed in the primary trial? Please clarify.

Comments from the reviewers:

Reviewer #1: This is a well conducted and well written study and the topic is both interesting and innovative in its use of a "user defined" end-point. The paper is given extra importance that the overall endpoint of the original study was neutral but the screening programme did reduce the combined end-point of MI and stroke, as chosen by the target population. This is very useful data to inform the design of potential future CV screening programmes. The fact that the adverse event of intra-cranial bleeding was also increased gives particular importance to the debate as to whether aspirin should be included in primary prevention programmes - even if screening has documented asymptomatic vascular disease. The authors do discuss this, but perhaps this section could be somewhat expanded given that there is real disagreement in different guidelines around the use of aspirin, although all guidelines concur on the use of statins.

Reviewer #2: This is an excellent piece of work examining the preferred patient outcome for CV screening and then the effects of screening with CT, ABBP and bloods on this outcome in Danish males. The effects of aspirin are salutary but those of statins seem impressive 

I have very few comments.

1. The assertion in line 2 that 'cardiovascular diseases remain the most frequent cause of death and is associated with a reduction in disability-adjusted life years.1' needs qualifying. In many parts of the world this is not the case and in some parts of Europe cancer is now more common.Perhaps 'cardiovascular diseases remain the most frequent cause of death in adults living in middle and high income countries'? 

2. At the end of the introduction the last sentence is unclear. 'In this study, we aimed to assess whether the comprehensive DANCAVAS screening is requested and leads to a reduction in a user-defined outcome'. Perhaps ..'aimed to assess the community interest in CV screening, patient preferred outcomes for such screening and the effects of screening on these outcomes. 

3. 'e-Boks' page 8 requires explanation for non Danish readers

4. 'at the time of postal' should read 'at the time of posting'. 

5. On page 12 Prevention of stroke and myocardial infarction was selected by 4,593 (50.5%) and 4,948 (54.4%), respectively, as the primary motivation for participating in the screening examination' underestimates the popularity of the end points. % Figures for those replying would be equally valuable and give a better insight in to the popularity of the end point. These could easily be added. The views on the preferred end point of those who didnt reply or who were not interested in screening anyway are of limited interest. 

6. The HRs for the outcomes are heavily dependent on the care offered as 'usual care'. Can the authors provide any data on how well or how badly CV risk is addressed in Danish primary care? Are men of this age routinely offered risk assessment using Q-risk or similar Framingham type assessments of risk? What is the ppn of Danish men of this age being treated with cholesterol and BP lowering drugs? The better this 'usual' care, the more difficult it will be to detect the effect of screening...

7. How were men with existing CVD handled? Were they included in DANCANVAS in which case the effects of the intervention will be diluted. 

Reviewer #3: Alex McConnachie, Statistical Review

Diederichsen et al present a secondary analysis of the DANCAVAS trial in which men were randomly assigned to receive the offer of a detailed cardiovascular screen or not. This paper is looking at a composite outcome of incident MI or stroke, since these two outcomes were selected as being of most importance to the target population. The choice of primary outcome is a clear strength of this paper, giving primacy to the outcomes that patients themselves are most keen to see prevented. This review considers the use of statistics in the paper.

Generally, I think these are very good. Main analyses are done on an intention to screen basis; there is a separate per-protocol analysis, though not a naive analysis, but done using inverse probability weighting. Outcomes are compared using Cox regression (with the proportional hazards assumption appropriately checked), and the number needed to invite (NNI) for screening to prevent one event is reported. Various secondary and explanatory outcomes are reported, in addition to the patient-selected primary outcome. I do have a few comments, but these are all quite minor.

The analyses use Cox models without adjustment, though one could argue that analyses should adjust for geographic region, since this was included as a stratification variable in the randomisation.

The statistical methods say that both relative and absolute risk estimates are reported, though I did not see anything in the results in terms of absolute risk differences. I think this would help when judging the benefit of screening in relation to the primary outcome, versus the increase in intracranial bleeding. The NNI is related to absolute risk differences, but it is not clear over what time frame these NNI estimates relate to. Note, the methods section talks of the NNI to save one life, even though this paper is reporting mainly non-fatal events as outcomes.

There is also mention of RMST analysis in the methods section, but again, I saw nothing in the results about this. RMST was to be used if there was evidence of non-PH in the Cox models. Perhaps this never occurred? If so, maybe this does not need to be mentioned in the methods section. I do not recall seeing anything about the PH assumption in the results. Perhaps this could be clarified.

This IPTW analysis is good to see, though the reference provided (number 10) is for the use of IPTW in observational studies. The only good reference I have found on the use of IPTW in randomised trials is Jo and Stuart (2009) (https://doi.org/10.1002/sim.3669), which describes how to estimate the CACE (complier average causal effect), which I think is what is being estimated in this study. They suggest using the odds of compliance (screening) to weight the control group, rather than the inverse probability of not being screened, which I believe is the same as using stabilised weights in standard IPTW, which may be slightly better. Also, in Jo and Stuart (2009), it is clear that non-compliers in the intervention group are excluded from the analysis (given a weight of zero) - that is not explicitly stated in the current paper, though I believe it was done.

Reviewer #4: Thank you for this very important study, which is part of the larger DANCAVAS study. The manuscript is written in a clear, concise and informative manner. I have no major comments to make, but I just wanted to ask whether your data would allow you to identify or suggest ways to adjust for the slightly increasing risk of intracranial haemorrhage with intensive medical prophylaxis (which is what you have found in this post-hoc analysis)? However, I realise that this is probably beyond the scope of the paper presenting the results of this study, and I congratulate you on your important work.

[LINK]

Comments from the academic editor:

I am familiar with the DANCAVAS trial. A few comments and questions:

1. The authors write that the user-defined outcome "was defined as a composite of reasons chosen by more than 50% of the population." Was this prespecified before the survey was done or post-doc? I think this is important to describe to readers because it was already known from DANCAVAS that the composite from stroke/MI/death has a p-value of <0.05. So there could have been an unintentional temptation ("researcher degrees of freedom) to develop a user outcome that was a "significant" result.

2. I would like more details on sampling of the target group. Were the men sampled using the same eligibility criteria and underlying registry from which the original participants were sampled? What is e-Boks? How do they know the men were not aware of DANCAVAS? What were the characteristics of men invited in the survey and how did those who participated differ from those who did not participate?

3. In addition to the IPTW analysis, I wonder if a useful complementary method to get at the effect size of screening itself (rather than the invitation to screen) would be to use an instrumental variable approach. Some experts use the phrase "Contamination adjusted intention to treat" for this approach; see the below citation. This isn't a requirement for me for publication, but I do wonder if it would strengthen the paper in giving clinicians an answer to the question that is most directly relevant to them, i.e., what is the effect size and NNT of screening for this patient in front of me.

Sussman JB, et al. BMJ 2010;340:c2073.

Cuzick J, et al. Statistics in Medicine 1997;16:1017-29.

This approach has been used a several important screening/invitation trials recently, for example, the recent colonoscopy trial in NEJM:

Bretthauer M, et al. N Engl J Med 2022;387:1547-56.

4. One thing that gives me pause is how the user-centered outcome was chosen.

The justification and methods feel very slender. There are lots of different consensus and deliberation methods to work with patients; why not choose one of those methods? Why are no other studies cited that build a background texture for this approach (i.e., like a formative paper or something, even an a priori protocol). How did theory guide this? It just feels a bit ad hoc as written currently.

This user-centered outcome is really the thing that advances science, yet there are minimal details on this. The outcome data has essentially already been published in the NEJM where MI/CVD/death was already reported as a "significant" secondary outcome (p. 7 of the NEJM paper).

1. Please upload any figures associated with your paper as individual TIF or EPS files with 300dpi resolution at resubmission; please read our figure guidelines for more information on our requirements: http://journals.plos.org/plosmedicine/s/figures. While revising your submission, please upload your figure files to the PACE digital diagnostic tool, https://pacev2.apexcovantage.com/. PACE helps ensure that figures meet PLOS requirements. To use PACE, you must first register as a user. Then, login and navigate to the UPLOAD tab, where you will find detailed instructions on how to use the tool. If you encounter any issues or have any questions when using PACE, please email us at PLOSMedicine@plos.org.

3. We ask every co-author listed on the manuscript to fill in a contributing author statement, making sure to declare all competing interests. If any of the co-authors have not filled in the statement, we will remind them to do so when the paper is revised. If all statements are not completed in a timely fashion this could hold up the re-review process. If new competing interests are declared later in the revision process, this may also h

---

## [Decision Letter · Decision Letter 2]

20 Mar 2024

Dear Dr. Diederichsen,

Thank you very much for re-submitting your manuscript "User-defined outcomes of the Danish Cardiovascular Screening (DANCAVAS) trial: a post hoc analyses of a population-based, randomised controlled trial" (PMEDICINE-D-24-00161R2) for review by PLOS Medicine.

I have discussed the paper with my colleagues and the academic editor and it was also seen again by 2 reviewers. I am pleased to say that provided the remaining editorial and production issues are dealt with we are planning to accept the paper for publication in the journal.

[LINK]

We look forward to receiving the revised manuscript by Mar 27 2024 11:59PM.   

Kind regards,

Pippa

Philippa Dodd, MBBS MRCP PhD

PLOS Medicine

plosmedicine.org

pdodd@plos.org

Requests from Editors:

GENERAL

Thank you for your very detailed and considered responses to previous editor and reviewer comments. Please see below for further comments which we require that you address in full. 

Many of the requirements pertain to formatting and some may not apply or may have been incorporated adequately already but please check ensure that each item is included as necessary.

DATA AVAILABILITY STATEMENT

Please provide a URL for inquiries to ‘Danish Statistics’.

STATISTICAL REPORTING

When reporting p values please report as <0.001 and where higher as p=0.002, for example. Please check and amend throughout all sub-sections of the manuscript and supporting files as relevant.

ABSTRACT

Please remove all current subheadings and revise as follows ensuring that your abstract reads as continuous prose. 

Suggest referring to the DANCAVAS trial in the past tense – ‘was’ as opposed to ‘is’.

Please structure your abstract using the PLOS Medicine headings (Background, Methods and Findings, Conclusions).

Please combine the Methods and Findings sections into one section, “Methods and findings”.

Abstract Background: Please provide context of why the study is important. The final sentence should clearly state the study question. The ‘user-defined’ outcome is a really novel part of this study and it would be worth elaborating a little on exactly what this means, for those less familiar. A reference to ‘patient preferences’ as per your introduction might be helpful.

Page 3 line 3 – please revise the phrase ‘reduce a user-defined outcome’ and be explicit as to what your study investigates, ‘…impacts an outcome determined by patient preferences…’ perhaps instead.

Please also replace ‘elderly’ with ‘older’ or simply state the age range without applying vintage to it. Please check and amend throughout all subsections of the manuscript and supporting files where relevant.

Abstract methods and findings:

Please ensure that all numbers presented in the abstract are present and identical to numbers presented in the main manuscript text.

Page 4 line 15 – please quantify with 95% CIs and p values.

Throughout, when reporting p values please report as <0.001 and where higher the exact p value as p=o.oo2, for example. 

Please include any important dependent variables that are adjusted for in the analyses.

Please include the actual amounts and/or absolute risk(s) of relevant outcomes (including NNT or NNH where appropriate), not just relative risks or correlation coefficients. (example for absolute risks: PMID: 28399126). 

Please include a summary of adverse events if these were assessed in the study.

In the last sentence of the Abstract Methods and Findings section, please describe the main limitation(s) of the study's methodology.

Abstract Conclusions:

Please address the study implications without overreaching what can be concluded from the data; the phrase "In this study, we observed ..." may be useful.

Please interpret the study based on the results presented in the abstract, emphasizing what is new without overstating your conclusions.

Please avoid vague statements such as "these results have major implications for policy/clinical care". Mention only specific implications substantiated by the results.

Please avoid assertions of primacy ("We report for the first time....") which can be risky!

AUTHOR SUMMARY

At this stage, we ask that you include a short, non-technical Author Summary of your research to make findings accessible to a wide audience that includes both scientists and non-scientists. The authors summary should consist of 2-3 succinct bullet points under each of the following headings:

• Why Was This Study Done? Authors should reflect on what was known about the topic before the research was published and why the research was needed.

• What Did the Researchers Do and Find? Authors should briefly describe the study design that was used and the study’s major findings. Do include the headline numbers from the study, such as the sample size and key findings. 

• What Do These Findings Mean? Authors should reflect on the new knowledge generated by the research and the implications for practice, research, policy, or public health. Authors should also consider how the interpretation of the study’s findings may be affected by the study limitations. In the final bullet point of ‘What Do These Findings Mean?’, please describe the main limitations of the study in non-technical language.

Author Summary should immediately follow the Abstract in your revised manuscript. This text is subject to editorial change and should be distinct from the scientific abstract. Please see our author guidelines for more information: https://journals.plos.org/plosmedicine/s/revising-your-manuscript#loc-author-summary

INTRODUCTION

Please address past research and explain the need for and potential importance of your study. Indicate whether your study is novel and how you determined that. If there has been a systematic review of the evidence related to your study (or you have conducted one), please refer to and reference that review and indicate whether it supports the need for your study.

Page 6, line 7-8 – ‘is, so far, the only…’ as above, assertions of primacy can be risky, suggest tempering, ‘to our knowledge’ may be helpful.

METHODS and FINDINGS

We agree with the statistical reviewer (please see below) regarding the inclusion of risk differences in table 3 and further details of the propensity score modelling.

Page 7, line 5 – as above, please replace ‘elderly’ with ‘older’ or simply state the age range without applying vintage to it. Please check and amend throughout all subsections of the manuscript and supporting files where relevant.

Page 7, lines 11-12 – thank you for citing your protocol and analysis plan. Please include the original study protocol document and analysis plan, with any amendments, as Supporting Information to be published with the manuscript.

Please also clarify the term ‘background population’ as used here.

Page 8, line 5 – please present numerators and denominators used to derive percentages and elsewhere as relevant, please check and amend throughout.

Page 8, line 18 – Suggest, ‘The target group included men…’

Page 8, line 20 – suggest, ‘…not aware of…’

Page 8, line 21 – please revise to ‘…was sent by…’

Page 9, line 1 – suggest ‘willingness to participate’

Page 9, line 4 – suggest, ‘was completed’

Page 9 – outcomes – thank you for helping to clarify our previous points. I wasn’t able to access the SAP for some reason (I think a problem with the site) but your point-by-point response suggests that these are different secondary outcomes as compared to the primary trial but what isn’t clear is how/why these were selected. Please include details.

Page 9, line 19 – suggest, ‘exploratory outcomes’

Page 9, line 21 – for clarity, please provide (brief) rationale for the choice of safety outcomes, in particular incident cancer (presumably radiation from CT imaging?)

Page 10, line 2 – please replace ‘send’ with ‘sent’.

Page 10, line 17 – please revise as follows, ‘…after six years of follow-up, estimated using Newcombe’s method’.

Page 11, line 2 – ‘the term probably-attendee-to-screening’ is difficult to understand and not well described. Please revise for clarity.

Line 12, please replace ‘done’ with ‘performed’.

Page 12, lines 3-4 – suggest, ‘6,182 (68.0%) responded’.

Page 12, line 17 – please revise to read as follows, ‘The relationship between…’

Page 12, line 20 – what do the percentages refer to here? Presumably percentage of the population affected. Please clarify.

Page 13, line 9 – please revise to read as ‘Exploratory Outcomes’

TABLES 

Please ensure that p values are reported as <0.001 and where higher the exact p value.

Please ensure that all tables are affiliated to an appropriate caption which clearly describes their content with out the need to refer to the text.

Please ensure that all abbreviations have been clearly defined in the caption/footnote/column header including those used for statistical reporting – HR, CI, IQR, etc.

Table 3 – we agree with the statistical reviewers comments please revise accordingly.

FIGURES

Please see here for guidelines on submitting and citing figures https://journals.plos.org/plosmedicine/s/figures#loc-how-to-submit-figures-and-captions

Please ensure that all figures are affiliated to an appropriate title and caption which clearly describes the figure content without the need to refer to the text.

Please ensure that all abbreviations are clearly defined an appropriate footnote or caption including those used for statistical reporting. 

Throughout please consider avoiding the use of green and/or red to make your figures more accessible to those with color blindness.

DISCUSSION

Please ensure that the discussion is structured as follows: a short, clear summary of the article's findings; what the study adds to existing research and where and why the results may differ from previous research; strengths and limitations of the study; implications and next steps for research, clinical practice, and/or public policy; one-paragraph conclusion. When revising please remove/avoid the use of sub-headings such that the discussion reads as continuous prose.

Page 18, lines 14 and 20 please remove these statements from the end of the discussion and include only in the manuscript submission form when you resubmit your manuscript. These will be compiled as metadata at the time of publication.

REFERENCES

For in-text reference callouts please place citations in square parentheses separate by commas. For example, [1,3,6] or [1-3]. Please check and amend throughout all sub-sections of the manuscript and supporting files.

In the bibliography please ensure that you list up to but no more than 6 author names followed by et al.

For all web references please ensure you include an, ‘Accessed [date].’

Journal name abbreviations should be those listed in the National Center for Biotechnology Information (NCBI) databases.

SUPPORTING INFORMATION

Please include the study protocol document and analysis plan, with any amendments, as Supporting Information to be published with the manuscript.

Please cite your Supporting Information as outlined here: https://journals.plos.org/plosmedicine/s/supporting-information

In the published article, supporting information files are accessed only through a hyperlink attached to the captions. For this reason, you must list captions at the end of your manuscript file. You may include a caption within the supporting information file itself, as long as that caption is also provided in the manuscript file. Do not submit a separate caption file.

Please ensure that all guidance detailed above for tables and figures in the main manuscript is applied to the supporting information.

SOCIAL MEDIA

To help us extend the reach of your research, please detail any X (formerly Twitter) handles you wish to be included when we tweet this paper (including your own, your coauthors’, your institution, funder, or lab) in the manuscript submission form when you re-submit the manuscript.

Comments from Reviewers:

Reviewer #2: An already very good manuscript has been improved b

---

## [Editor Report · Decision Letter 3]

12 Apr 2024

Dear Dr Diederichsen, 

On behalf of my colleagues and the Academic Editor, Dr. David Flood, I am pleased to inform you that we have agreed to publish your manuscript "User-defined outcomes of the Danish Cardiovascular Screening (DANCAVAS) trial: a post hoc analyses of a population-based, randomised controlled trial" (PMEDICINE-D-24-00161R3) in PLOS Medicine.

Prior to publication please address the following revisions:

1) ABSTRACT

Page 3, line 1 – please amend to read as follows, ‘The Danish cardiovascular screening (DANCAVAS) trial, a nationwide trial designed to investigate the impact of cardiovascular screening in men, did not…’

Page 4, line 3 – please amend to read as follows ‘The primary limitation of the study is that exclusive enrolment…renders…’

Page 14, line 16 – please remove ‘Tweet’ we have recorded this for you.

2) AUTHOR SUMMARY

Please revise as follows for improved grammar and accessibility.

Why Was This Study Done? 

• Stroke and heart attack remain a prevalent cause of decreased quality of life and premature death.

• Coronary atherosclerosis serves as an important risk modifier and is easily identifiable by cardiac imaging.

• Despite this, screening for cardiovascular disease by cardiac CT in the Danish cardiovascular screening (DANCAVAS) trial, did not lead to a decrease in death from cardiovascular causes.

• However, it is essential to recognize that patients may have preferences other than death, and these preferences play a critically important role in informed and shared decision-making between caregivers and patients.

Line 16 – please replace ‘aneurisms’ with ‘aneurysms’.

Line 17 – please revise to read as follows, ‘Here, the study outcomes were determined…’ as currently written the text suggests that for the primary trial outcomes were determined by patient preference.

Line 21 – suggest, ‘intracranial haemorrhage’.

What do these findings mean?

Line 2 – please replace, ‘meet’ with ‘met’.

Line 3 – please replace ‘brain’ with ‘intracranial’.

Line 4 – please revise as follows, ‘These data suggest that there is limited benefit to the widespread implementation of cardiovascular screening programmes.’

It would be helpful to include an additional point explaining what needs to happen next. Can we improve CV screening programmes? Should we focus on screening differently? Or should we focus on healthy life-style interventions instead? Or a combination of the above? Please be brief but clear.

3) INTRODUCTION

Line 3 – please change ‘is’ to ‘are’.

4) TABLES 

Table 2 & 3 – incidence risk difference column. To improve reader accessibility it would be helpful to place the 95% CIs below the RD to prevent splitting across rows. 

It seems evident only where you present negative values. You could reduce the size of column 1 to give you more space.

Please also revise as relevant in the supporting information tables – for example supplementary table 2.

5) DISCUSSION 

Page 16, Line 15 – note the citation in superscript – please follow referencing guidance detailed below (and throughout).

Page 17, line 6 – sentence beginning ‘Aspirin…’ please revise to read as follows, ‘Prescribing aspirin to…’

Page 17, line 12 – please define ‘MESA’ apologies if I have missed it.

Page 17, line 21 – please replace ‘diluted’ with ‘attenuated’.

Page 18, line 18 – please replace ‘restrict’ with ‘Thus restricting external validity and generalisability’.

Page 18, line 21 – it would be helpful to (briefly) define the ‘flaws’ you refer to here.

Page 19, line 3 – please remove the sub-heading ‘Conclusion’.

Page 19, line 6 – as for the author summary above, a point about what could or should be done next to move things forward would be helpful.

Page 19, line 7 – please replace ‘jet’ with ‘yet’.

6) REFERENCES

For in-text reference callouts please place citations in square parentheses separate by commas and preceding punctuation. For example, ‘[1,3,6].’ or ‘[1-3].’ Please check and amend throughout all sub-sections of the manuscript and supporting files.

PRESS

Thank you again for submitting to PLOS Medicine, it has been a pleasure handling your manuscript. We look forward to publishing your paper. 

Kind regards,

Pippa

Philippa C. Dodd, MBBS MRCP PhD 

PLOS Medicine

pdodd@plos.org